# Human Cytomegalovirus Oncoprotection across Diverse Populations, Tumor Histologies, and Age Groups: The Relevance for Prospective Vaccinal Therapy

**DOI:** 10.3390/ijms25073741

**Published:** 2024-03-27

**Authors:** Marko Jankovic, Tara Knezevic, Ana Tomic, Ognjen Milicevic, Tanja Jovanovic, Irena Djunic, Biljana Mihaljevic, Aleksandra Knezevic, Milena Todorovic-Balint

**Affiliations:** 1Department of Virology, Institute of Microbiology and Immunology, 1 Dr Subotica Street, 11000 Belgrade, Serbia; aleksandra.knezevic@med.bg.ac.rs; 2Faculty of Medicine, University of Belgrade, 8 Dr Subotica Street, 11000 Belgrade, Serbia; tara.knezevic@gmail.com (T.K.); anatomic9977@gmail.com (A.T.); ognjen011@gmail.com (O.M.); irenadjunic04@gmail.com (I.D.); bilja.mihaljevic@gmail.com (B.M.); bb.lena@gmail.com (M.T.-B.); 3Institute of Medical Statistics and Informatics, 15 Dr Subotica Street, 11000 Belgrade, Serbia; 4Institute for Biocides and Medical Ecology, 16 Trebevicka Street, 11000 Belgrade, Serbia; tanja.jovanovic@biocidi.org.rs; 5Clinic of Hematology, University Clinical Centre of Serbia, 2 Dr Koste Todorovica Street, 11000 Belgrade, Serbia

**Keywords:** cytomegalovirus, oncogenesis, oncoprotection, cancer, global, T cell

## Abstract

The oncogenicity of the human cytomegalovirus (CMV) is currently being widely debated. Most recently, mounting clinical evidence suggests an anti-cancer effect via CMV-induced T cell-mediated tumor destruction. However, the data were mostly obtained from single-center studies and in vitro experiments. Broad geographic coverage is required to offer a global perspective. Our study examined the correlation between country-specific CMV seroprevalence (across 73 countries) and the age-standardized incidence rate (of 34 invasive tumors). The populations studied were stratified according to decadal age periods as the immunologic effects of CMV seropositivity may depend upon age at initial infection. The International Agency for Research on Cancer of the World Health Organization (IARC WHO) database was used. The multivariate linear regression analysis revealed a worldwide inverse correlation between CMV seroprevalence and the incidences of 62.8% tumors. Notably, this inverse link persists for all cancers combined (Spearman’s ρ = −0.732, *p* < 0.001; β = −0.482, *p* < 0.001, adjusted R^2^ = 0.737). An antithetical and significant correlation was also observed in particular age groups for the vast majority of tumors. Our results corroborate the conclusions of previous studies and indicate that this oncopreventive phenomenon holds true on a global scale. It applies to a wide spectrum of cancer histologies, additionally supporting the idea of a common underlying mechanism—CMV-stimulated T cell tumor targeting. Although these results further advance the notion of CMV-based therapies, in-depth investigation of host–virus interactions is still warranted.

## 1. Introduction

The human cytomegalovirus (CMV), a widespread and globally prevalent agent of infection, affects approximately 83% of the world’s population [1]. However, the extent of viral infection varies considerably worldwide, with seroprevalence reaching up to 100% in specific populations [2]. Following initial exposure, CMV establishes a lifelong infection within the host, typically without causing symptoms. This is sharply contrasted with the serious illness it can provoke in individuals with compromised immunity.

Although not categorized as an oncogenic virus, this pathogen has been linked to a wide array of cancers [3,4,5,6], and the debate over whether it possesses oncogenic potential has been extensive. Over the past few decades, however, an accumulating body of evidence suggests that the virus may, in fact, exhibit anti-tumor properties [7,8,9,10,11,12,13]. Recently, and for the first time, this phenomenon was observed on a global scale in malignancies of the B-cell lineage [7]. The underlying rationale for this type of anti-cancer behavior is rooted in a virus-focused immune response, where CMV molecules expressed on tumor cells serve as target antigens. For quite some time, researchers have been exploring viral antigens present within cancer cells as promising candidates for tumor-specific cancer immunotherapy [12]. These approaches encompass strategies such as cytotoxic T-lymphocyte (CTL)- or dendritic cell (DC)-based vaccines. The existing evidence substantiates the idea that CMV-derived antigens can function as potent inducers of immune responses against infected cancer cells.

Despite the number of single-center and in vitro investigations conducted thus far, there have been no comprehensive studies exploring the potential anti-tumor influence of CMV on a global perspective. Additionally, the question of whether an individual’s age at the time of first infection plays a significant role in the generation of oncoprevention has yet to be resolved. In this work, we aimed to examine the connection between CMV serodispersion and the rate of occurrence of widely prevalent and histologically variegated tumors at a global scale. Also, we seek to determine whether CMV anti-cancer effects, if any, show consistencies over distinct age groups. Calculating correlations according to age-stratified groups allows for a rough spectral resolution of data that covers the evolution of virus prevalence in an entire population, as we continue to observe the signal travelling by 10-year bandwidths in time. Building upon previous similar research, we present an outlook on CMV as a plausible agent for preventing the onset of cancer across various demographic, geopolitical, and socio-economic strata.

## 2. Results

An inverse Spearman’s correlation was evident between the prevalence of CMV and 88.2% (30/34) of the estimated age-adjusted tumor incidence rates, demonstrating a high degree of statistical significance (*p* < 0.001) in 73.5% (25/34) of instances (Table 1). Of note is the persistence of this statistical association when considering incidence rates for all cancers combined (Spearman’s ρ = −0.732, *p* < 0.001; shown in Figure 1). This observation suggests a plausible protective influence of the virus against the aforementioned neoplastic conditions on a global scale.

Strikingly, no discernible correlation surfaced between the CMV pervasiveness and the frequency of Kaposi’s sarcoma (Spearman’s ρ = −0.007, *p* = 0.953; shown in Figure 2). This further supports the oncopreventive faculty of CMV, as it is in line with the hypothesized CMV-galvanized T-cell tumoricidal activity; namely, the T-cell immune response is severely impaired in individuals afflicted by HIV/AIDS, who prominently present with Kaposi’s sarcoma, so it cannot be mobilized with adequate efficacy against tumors.

Conversely, CMV was significantly and positively correlated with nasopharyngeal carcinoma (Spearman’s ρ = 0.226, *p* = 0.023; shown in Figure 3) and gallbladder cancer (Spearman’s ρ = 0.316, *p* = 0.006), which constitutes 5.88% of studied tumors. All malignancies and their corresponding association with CMV are represented in Table 1.

Socioeconomic status (SES) influences CMV prevalence. An analysis of the disparity in CMV prevalence and overall cancer incidence according to the country’s income level was performed (Figure 4), as affluence is an integral part of the SES of a population. The results showed a significant difference in the given parameters between the tested groups (*p* < 0.001). Of note, the incidence of cancer is significantly greater in high-income nations (*p* < 0.001). Moreover, the prevalence of CMV is comparatively lower in these countries compared to middle-income (*p* < 0.001) and low-income states (*p* = 0.015). No statistically significant difference in CMV prevalence was observed between middle- and low-income countries.

A univariate linear regression (ULR) analysis was performed using CMV as an independent factor (Table 1). About three quarters (25/34, 73.5%) of tumors showed a significant association to CMV prevalence. Additionally, the analysis attained significance for the incidence of all cancers combined. Cytomegalovirus prevalence presented as a significant independent predictor for tumor incidence, with the highest negative standardized β values for kidney tumors (β = −0.792, *p* < 0.001), meaning that an increase in the independent variable (CMV prevalence) is highly associated with a decrease in the dependent variable (specific tumor incidence). CMV prevalence was a good independent predictor in the univariate linear regression model, with high adjusted R^2^ coefficients (Appendix A), meaning that high percentages of the variation of incidence for given tumors could be explained with the univariate regression model. Additionally, CMV prevalence was a significant, although positive, predictor for tumors of the gallbladder (β = 0.265, *p* = 0.023), cervix (β = 0.234, *p* = 0.046) and borderline nasopharynx (β = 0.227, *p* = 0.053), respectively. 

A multiple linear regression (MLR) analysis was performed with several confounding factors (Cf. Materials and Methods). Out of all the tested parameters, measured CMV prevalence adjusted for HDI as a confounding factor was the best-fitted model to present the association of CMV with tumor incidence. Multiple linear regression (MLR) analysis confirmed the antithetical association of CMV prevalence with tumor incidence in a majority (25/34, 73.5%) of tumor localizations (Table 1). The adjusted R^2^ coefficients were the highest for kidney tumors (R^2^ = 0.771, Adj. R^2^ = 0.765), which means that 76.5% of the variation for kidney tumors could be explained with this model (Appendix A). However, certain tumors’ incidences (hypopharynx, thyroid, ovary, cervix and penis) showed no association to CMV prevalence when adjusted for HDI. Finally, after adjusting for other aforementioned confounding factors, the results of the MLR analysis did not change significantly. For details concerning R^2^ values and confidence intervals (CI), consult Appendix A.

The detailed examination of the relationship between CMV prevalence and worldwide cancer rates in specific age groups is presented in Table 2. The significant and opposite correlation observed at the country level manifests in specific age brackets for 94.1% (32/34) tumors. Notably, this link tends to be more pronounced in older age groups for most cancer types, with exceptions such as Kaposi’s sarcoma, cervical carcinoma, and malignant liver dyscrasia, which predominantly exhibit a protective effect in individuals under 40 years old. Finally, the correlation is still highly significant and speaks in favor of oncoprotection for all tumors combined, regardless of the age group.

A positive correlation in the case of nasopharyngeal carcinoma and gallbladder cancer is observed once again, suggesting the oncogenic potential of CMV in these types of cancers. Notably, the virus’s pro-tumor impact on individuals with gallbladder cancer is significantly more prevalent between the ages of 20 and 70. A similar pattern is discernible for cervical carcinoma, which is linked to individuals aged over 50.

In a smaller number of cases (23.3%), CMV appears to have an association with the development of tumors in specific age brackets, even in cases where CMV generally correlates with cancer prevention, when examining combined incidence rates. It is noteworthy that here, the pro-tumor effect becomes apparent at a younger age, which is in stark contrast to the oncoprotective effect observed in older individuals. The neoplasia where this is the case are those of the hypopharynx, larynx, lip/oral tumors, liver, oropharynx, esophagus, stomach, and multiple myeloma.

## 3. Discussion

To date, there have been various studies both supporting and challenging the idea of CMV oncogenicity. Our research aimed to provide further perspectives on this issue by presenting a global viewpoint. Additionally, we have explored the potential time-dependent effect of CMV infection in a variety of tumor histologies.

### 3.1. Cytomegalovirus and Oncoprotection—From T-Cells to Vaccines

Characterizing CMV solely as an agent of oncogenesis is an outdated concept. Accumulating evidence not only supports its role in oncomodulation, but also points towards its potential for oncoprotection [7,9,10,14,15].

Recently, the perspective of CMV functioning as a safeguard against cancer has gained attention. Notably, recent clinical studies have reported apparent anti-tumor effects of CMV in individuals with colorectal cancer and bronchogenic carcinoma [9,10]. Furthermore, patients with B-cell malignancies were noted to have a significantly lower incidence of CMV seropositivity compared to the control group [7]. A conspicuous absence of human CMV DNA was observed in studies reviewing pleomorphic adenomas [16], Warthin’s tumors [16], epithelial ovarian cancer [17], papillary thyroid cancer [18], pediatric medulloblastomas [19], and central nervous system tumors [20]. Based on histological and clinical data in the case of cervical cancers, Thompson et al. reported that there is no substantiated evidence to suggest a connection between CMV-positive cancers and any atypical histologic cell types or a more aggressiveness in clinic [21].

In patients who have undergone allogeneic hematopoietic cell transplantation (HCT), CMV reactivation has been associated with a marked reduction in leukemia relapse risk [8]. This is corroborated by the observation that prompt CMV replication may mitigate risk of relapse in non-Hodgkin lymphoma [22], acute myeloid leukemia [23,24,25], and pediatric acute leukemia in the wake of HCT [26]. In a cohort of patients with myeloproliferative disorders, reactivation of CMV following HCT was linked to a slight decrease in the risk of early relapse [27]. A similar association was reported in solid organ transplant (SOT) patients; Geris and colleagues concluded that CMV status did not correlate with the risk of developing secondary cancer in SOT recipients [28]. Moreover, these authors acknowledge an inverse correlation between CMV and diffuse large B-cell lymphoma (DLBCL), which is in accordance with our earlier study [7]. A comparable outcome was noted in an experimental model involving murine CMV, where the virus negatively influenced the progression of B-cell lymphoma [29].

In the current work, univariate linear regression provided a significant and inverse link between CMV seroprevalence and tumor incidence for 73.5% (25/34) of the malignancies studied (Table 1). After performing a multivariate regression analysis and adjusting for the human development index (HDI) (as a proxy for socioeconomic status) closely associated with CMV infection, the statistically significant antithetic link was still upheld for 62.8% of the investigated cancers (22/34; Table 1). Furthermore, the analysis confirms that CMV is the sole significant factor for oncoprotection in tumors of the salivary glands, lip/oral cavity, oropharynx, vulva and non-melanoma skin cancers. Herein, the HDI does not seem to partake in tumor incidence variation. The significant link noted using the univariate regression was lost following the multivariate analysis for tumors of the thyroid, ovary, uterine cervix, penis, and hypopharynx. This hints at CMV as not conferring protection in these cancers. The reason for the discrepancy in these tumors is not understood. It could be potentially linked to a reduced viral affinity for these specific histological environments. Finally, no significant association for cancers of the larynx, esophagus, stomach, Kaposi’s sarcoma, or liver was confirmed by our ULR and MLR.

Conversely, both HDI and CMV show significant associations in another complement of malignancies: melanoma, kidney, breast, testis, colorectum, prostate, corpus uteri, pancreas, multiple myeloma, leukemia, Hodgkin’s disease, non-Hodgkin lymphoma, mesothelioma, lung, brain/CNS, and the bladder. As the CMV prevalence increases, the incidence of cancers decreases; the opposite is noted for HDI, which is co-incremental with tumor pervasiveness. The increase in HDI within economically thriving communities corresponds to the dissemination of improved hygiene, sanitation, education, and overall better health practices as compared to their less prosperous counterparts. Consequently, this trend could result in reduced transmission and seroprevalence of CMV, thereby diminishing its potential oncopreventive impact. Such a connection would additionally reinforce the comprehensive anti-cancer characteristics of CMV infection observed both across various tumor types and the world over.

The univariate regression model yielded CMV prevalence as a significant and positive predictor for gallbladder and nasopharyngeal tumors. This pro-tumor effect remains after employing the multivariate analysis, suggesting that CMV is a potential oncogenic agent in these malignancies.

A novel association (*p* = 0.007) emerged for vaginal tumors, whose frequency decreased in concert with increases in both CMV prevalence and HDI. This suggests a protective influence conferred by CMV, but also places less economically prosperous countries at an advantage over vaginal cancerogenesis. The inverse correlation between HDI and vaginal cancer is an intriguing one and may be explained by specific exposures or behavioral habits characteristically present in wealthier countries.

Affluence is a key component of the socioeconomic status (SES) of a population, and SES is known to impact CMV prevalence [30,31,32]. Our study reveals a notable increase in cancer incidence in high-income nations (*p* < 0.001) and a lower cancer rate in middle- (*p* < 0.001) and low-income nations (*p* = 0.015) with a lower CMV prevalence. These findings suggest a reverse correlation between viral prevalence and tumor incidence as regards country wealth, indicating that CMV prevalence is higher and cancer frequency lower in economically disadvantaged countries. When considering HDI as a confounding variable in MLR analysis, high CMV prevalence still remained a significant predictor for lower cancer incidence of most tumors. MLR analysis showed CMV as statistically insignificant in relation to tumorigenesis in the thyroid, ovaries, penis, and hypopharynx. This suggests that HDI is a more influential predictor in explaining the observed variability in these particular tumor types, as compared to CMV, with higher HDI associated with higher malignancy incidences.

Over a century ago, speculation began to arise that certain viruses carry the capacity for tumor regression and remission [33]. It was found much later that these viruses possess an innate predilection for cancer cells, which both destroy the infected cells and set off host anti-cancer immunological mechanisms [34]. Additionally, oncolytic viruses, whose main purpose is to lyse tumor cells, have also been noted within coxsackievirus, adenovirus, and herpes simplex virus strains [35].

Recently, Ye et al. described an association between the highly conserved *US31* CMV gene and its role in suppressing tumor proliferation and metastasis [36]. Further manipulation of the tumor microenvironment is managed through the virus’ transformative effects on various cellular genes and signaling pathways [37,38,39,40]. In addition to its direct interactions with host genes, human CMV influences the overall genetic landscape of tumor cells [41]. This includes the promotion of apoptosis [37,40,42,43], influence on the production of cytokines and chemokines [39], and induction of a vigorous activation of immune cells that penetrate tumors [37]. Further evidence is provided by a delay in tumor growth by the primary infection of tumor nodules [44].

In terms of cellular modifications, murine CMV has been observed to both engage [45] and infect [38] macrophages at the tumor site, where they are modified to enhance antitumor immune responses and effectively hinder oncogenesis [40,45]. Also noted is an immune response that principally activates natural killer (NK) cells [41,46,47,48,49,50], followed by CD4+ and CD8+ cytotoxic T-lymphocytes [41,50,51,52], high-affinity antibodies [41], and enduring memory T-cells [50]. Other studies describe adoptive T-cell therapy, based around CMV-specific T-cells and dendritic cells pulsed with viral pp65 RNA as a means to vaccinate patients suffering from glioblastoma multiforme (GBM); promisingly, these works have reported on improved survival in GBM patients [53]. Caution must be exercised, however, as the success of CMV-based therapies in this tumor setting may not be extrapolated to other types of malignancies with apodeictic certainty.

Harnessing this concentrated immune system reaction, most recent preclinical and clinical studies regarding CMV as an anti-cancer vaccine have shown encouraging results [54,55]. Additionally, CMV promoters have proven to successfully govern p53 tumor suppressor gene therapy when used as a chemotherapeutic for human ovarian carcinoma [56]. The results yielded by our investigation hint that this oncoprotective effect conferred by CMV may stem from the very T-cell anti-tumor onslaught the vaccination efforts describe. Specifically, the hypothesized CMV-mediated tumoricidal activity is posited to be modulated through an intact T-cell immune response, which is significantly compromised in individuals afflicted by HIV/AIDS. In this work, no correlation between CMV oncoprotection and a cumulated incidence of Kaposi’s sarcoma was evidenced by either ULR or MLR. Considering that a fully functional T-cell repertoire is mandatory for anti-tumor activity, the absence of an association further bolsters the conjecture regarding CMV’s oncopreventive capacity.

Recently, murine-CMV-derived peptide epitopes appeared to be good “homing beacons” for a robust anti-tumor immune response, as suggested in a recent work by Çuburu et al. [57]; this work might then translate into human CMV research with viral proteins serving as quality biomarkers for vigorous T-cell responses. Research on targeting CMV as the biomarker suggests selective immunotherapy in the treatment of medulloblastoma [58], glioblastoma multiforme [59], as well as pancreatic cancer and brain tumors by the means of B-lymphocytes [60]. Immune recognition and elimination of CMV-infected tumor cells contribute to the anti-tumor properties attributed to the virus.

Cytomegalovirus is undoubtedly emerging as a strong anti-cancer vaccine candidate [61]. Although more controlled trials are required, we believe that this discovery of a worldwide putative oncopreventive effect of CMV can advance the notion of CMV as an agent of active immunization against a wide range of malignant diseases.

### 3.2. When Might CMV Provide the Highest Level of Protection?

Apart from this study, so far there has been no comprehensive or global research into possible time-dependent oncogenesis (or oncoprotection) linked with CMV infection. It is worthwhile noting, however, that congenital CMV infection correlated with the development of childhood blood cancer in certain studies [20,62], indicating that early infection might predispose for neoplastic events.

In our investigation, the purported oncoprotective effect of CMV had a propensity for older age groups for most cancer types. This suggests that an infection taking place at a later stage in life might offer protection against developing cancer, as opposed to an infection occurring earlier. Alternatively, it advocates for an onco-preventive effect that is more pronounced in tumors that develop in older populations. Nevertheless, this correlation remains robust when considering the cumulative incidence of all types of malignancies collectively. Finally, some tumors exhibit this contrasting correlation across all age groups, indicating a potential tumor-inhibiting effect irrespective of the timing of the initial infection.

The anti-cancer property notwithstanding, in certain age groups, CMV acts as a de facto agent of oncogenesis (Table 2). This is most obvious for nasopharyngeal carcinoma and gallbladder cancer. The link with NPC has already described in the literature [63], which further corroborates our findings from a global perspective. Interestingly enough, in specific cases (23.3%) of malignancies where CMV generally correlated with cancer prevention, in some age groups, its effect would be statistically recognized as oncogenic (Table 2). This pro-tumor agency was apparent in younger age groups.

The dual nature of CMV (pro- and anti-oncogenic) observed herein, that manifested more clearly when discrete age brackets were analyzed, may point to an outcome (tumor genesis or tumor prevention) predicated on the time of first infection. Evidence indicates that acquiring the infection earlier in life or congenitally increases the individual’s susceptibility to cancer development [20,62], whereas encountering CMV at a later stage may demonstrate some protective effects. However, the conclusions should be taken with caution, as correlation does not always confer causation (*Cf*. Study limitations).

### 3.3. Arguments in Favor of CMV Oncogenesis

To date, the role of CMV as a potential underlying factor of malignancies is still debated. Its oncogenic role has indeed been postulated numerous times [64,65,66], with the virus posited to be involved in over 90% of the most frequently presenting tumors [67]. CMV is purported to possess the ability to influence cellular processes and pathways, potentially increasing the cell’s susceptibility to developing malignancies by interfering with the cellular pathways associated with the cell cycle, apoptosis, angiogenesis, cell invasion, and the immune response of the host [68,69]. Furthermore, it has been proposed that CMV may promote tumor growth [61]. 

Polz-Gruszka et al. have detected CMV DNA in fresh-frozen tumor tissue fragments from 10% of patients with oral squamous cell carcinoma [70]. Cytomegalovirus has been reported as a risk factor for glioma, neuroblastoma, as well as breast cancer [3,4,71,72,73,74]. Persons suffering from breast cancer who were CMV-seropositive or had CMV DNA in tumor tissue were significantly more likely to develop Stage IV metastatic tumors, hinting at an adverse oncomodulatory role of the virus, which promotes metastases [75]. An adverse effect of CMV was also noted elsewhere regarding the same pathology [76]. Furthermore, viral protein expression was found to relate to shorter overall survival in patients with breast cancer [77]; CMV *IE2* gene expression was also associated with this tumor [78]. In a recent study by Paradowska and colleagues, a significant proportion (70%) of epithelial ovarian cancers (EOC) contained CMV DNA; moreover, the pathogen was significantly more prevalent in EOC than in benign tumors [79]. Evidence suggests that CMV is involved in the pathology of colorectal cancer (CrC) and inflammatory bowel disease [80,81,82,83,84]. Cytomegalovirus infection was also associated with a poor prognosis in CrC patients, where three viral genes (*UL82*, *UL42*, and *UL117*) were linked to poor patient survival outcomes [80]. The virus was postulated to play a role in the tumorigenesis of malignant gliomas [69,83,85,86,87,88,89,90,91,92,93], notably the extremely destructive glioblastoma multiforme (GBM) [68,94,95], although it is not considered to have a role in the development of non-GBM infantile brain tumors. Both CMV and Epstein–Barr virus (EBV) were detected in the exhaled breath condensate of lung cancer patients, and were consequently potentially implicated in lung carcinogenesis by the authors [96]. In patients with head and neck malignancies, CMV seropositivity itself was not found to impinge on survival; however, the authors propose that high titers and active CMV virus in the tumor environment may be linked to inferior outcomes [97]. The study conducted by Sarshari and colleagues found that CMV, EBV, and human herpesvirus 6 might play a role in the initiation and development of gastritis and gastric cancer [5]; the potential risk for gastric and gastrointestinal cancer was acknowledged elsewhere [74,98].

Cytomegalovirus has been also associated with prostate cancer [64,83,99], colon and cervical carcinoma [6,100,101], as well as epithelial ovarian cancer [102]. Recently, congenital CMV infection was put forward as a risk factor for childhood acute lymphoblastic leukemia [62]. CMV infections have been noted to cause chronic inflammatory processes, which in and of itself serves as both a precursor and cornerstone of malignancy [103,104,105,106,107,108,109,110]. The virus is known to directly seize tumor-promoting cellular events while simultaneously overriding immunosuppressive mechanisms [67]. 

At a minimum, CMV has a role in oncomodulation [111], as it supports the proliferation and longevity of cancer [112,113,114] while increasing its malignant potential by progressively inducing more neoplastic phenotypes [83,85,113]. This is achieved by means of large-scale derangement of cellular signaling pathways, disordered enzyme expression, and chronic inflammation [83,84,113], which may precipitate genomic injury [115]. However, it is worth noting that not all oncomodulatory effects need be detrimental to the host. Namely, it has been observed that CMV may inhibit the migration of specific breast cancer cells [116].

Some high-risk CMV strains were implicated to have a catalytic role in the explicit transformation of primary cells [112,117,118,119]. Most notably, Cobbs observed that CMV not only has the capacity for epithelial cell modification but is implicated in epithelial to mesenchymal (EMT) transformation in tumor cells and *vice versa*. It is important to note that EMT has been indicated as the causative agent of cell-to-cell adhesion loss, deranged cellular polarity and cytoskeleton transfiguration [120], hence facilitating a critical role in tumor progression and functioning as a primary target of interest in anticancer therapy [121]. Cytomegalovirus could present as a causative agent to GBM through ARG2 upregulation [86] and STAT3 signaling [68], which is often used as an early tumor biomarker. Chemokine receptor US28 binds and activates a proliferative response that is capable of promoting tumorigenesis [122]. CMV can cause sequestrations and deactivates p53, which is proving to be important in our understanding of the virus’ tumorigenic properties [123].

### 3.4. Study Limitations

We consider it highly significant to acknowledge the limitations inherent in our research. The country’s overall CMV prevalence, as obtained from Zuhair et al. [1], does not precisely mirror the virus distribution among age groups of a particular country. This results in a limited accuracy of CMV seroprevalence distribution within decadal samples of a population. Provision must be made for a possible gain in accuracy if the time intervals explored have been significantly shorter than a decade, thus diminishing the intervening range of separation between the age groups. Despite a rather large probable error imposed by the widths of successive decadal intervals, the CMV prevalence used could reasonably serve as a proxy, given that significant associations calculated for combined incidence rates generally remained consistent when compared to individual age-specific incidences. Consequently, even without geographic information, we allowed for a probabilistic mixture of prevalence while still having to account for correlations between them.

Furthermore, the correlation we used in the statistical analysis does not necessarily infer causation, although the inverse association between CMV prevalence and cancer age-standardized incidence rates is striking, as it covers a large proportion of neoplasia the world over. A number of factors contribute to CMV seropositivity: age [124,125,126], gender [124,127,128], socioeconomic status [30,31,32], current smoking [30], level of education [30], number of sexual partners [30], childcare practices [125], and different cultural conditions or customs related to breastfeeding [125], to name some of them. Racial/ethnic background is also related to SES [129,130]. However, the multivariate analysis places CMV as a variable that serves independently or in concert with HDI as an oncoprotective factor against a wide complement of malignancies worldwide; none of these factors appears to be as good an oncoprotective candidate as CMV appears to be. 

It is worth noting that EBV itself is a tumorigenic virus that has a predilection to cause nasopharyngeal carcinoma. Hence, we would advise exercising caution when interpreting our results that link CMV to these cancers, as a co-infection with EBV (acting as a confounding factor) might impinge on the veracity of our conclusions.

It is important to mention that data on the prevalence of CMV, as well as the Human Development Index (HDI), were attainable for a total of 73 and 72 nations around the globe, respectively. Furthermore, information regarding some of the other potentially confounding variables (estimated number of sexual partners, smoking prevalence, and breastfeeding) that we attempted include in the final MLR model were available for fewer than 72 countries in the multivariate analysis. Therefore, these findings may have probably yielded more precise results if data were provided for a greater number of countries.

Our conclusions do not claim to serve as the proverbial *smoking gun* in the effort to elucidate the role of CMV in malignancies. Instead, this study highlights a compelling correlation between this enigmatic pathogen and a wide range of neoplasms across diverse histological backgrounds and populations. Ultimately, a thorough understanding of the relationship between tumors and CMV is likely to emerge through extensive molecular analyses and prospective studies involving large population cohorts. 

## 4. Materials and Methods

To investigate the potential anti-oncogenic effects of CMV, we explored the correlation between country-specific age-standardized cancer incidence rates and corresponding CMV seroprevalences. In order to compensate for any confounding variables (i.e., those that might influence CMV pervasiveness), we employed multivariate logistic regression (MLR). 

The age-adjusted annual incidence rates (per 100,000 individuals) specific to 34 cancer categories were documented across 185 countries and were sourced from the Global Cancer Observatory (GLOBOCAN), a division of the World Health Organization [131]. Incidences were observed jointly for males and females, encompassing the full listed age range (0–85+ years).

The prevalence of CMV was depicted through country-specific viral seroprevalence data for a total of 73 countries. This information was gathered by Zuhair and colleagues [1], who conducted a systematic survey of the published literature to provide insights into the worldwide prevalence of CMV IgG antibodies. The list of investigated malignancies is presented in Table 1.

The aforementioned data sources were then used to inquire into the potential time-dependent relation between CMV infection and cancer. Namely, we asked whether there is a specific age range where the association between CMV and cancer incidence would appear more prominently. Incidences for all malignancies were subsequently disaggregated into 10-year age intervals. Known cancer incidence rates for each age interval were then compared with the corresponding country-specific CMV prevalence (Table 2).

In both cases, the comparison between age-standardized annual cancer incidence rates and country-specific CMV seroprevalence was statistically analyzed by using the Spearman’s rank correlation test. For non-temporal, aggregated data, Kruskal–Wallis analysis was used for variables without normal distribution followed by the Bonferroni–Dunn correction as a post hoc test. Univariate and multivariate linear regression analyses were performed for every tumor type/localization, using CMV as an independent predictor in the univariate analysis and adjusting for confounding factors in the multivariate analysis. Confounding factors were chosen based on their literature-based association with CMV prevalence: the Human Development Index (HDI) was used as a parallel of socioeconomic status [30,31,32], average population age [124,125,126], estimated number of sexual partners [30], smoking prevalence (as a stand-in for current smoking) [30] and the percentage of children born in the last 2 years who were ever breastfed [125]. The *p*-value was accepted as <0.05 for the statistical significance level. The information was provided by the United Nations Development Program 2021 Human Development Reports [132], United Nations World Population Prospects (2022) [133], the World Population Review website [134], the Tobacco Atlas (University of Illinois, WHO GTCR 2023 data) [135] and UNICEF [136], respectively. Finally, to inquire into the country income level, data from the World Bank were used [137]; note that “for the current 2024 fiscal year, low-income economies are defined as those with a GNI per capita, calculated using the World Bank Atlas method, of $1135 or less in 2022; lower middle-income economies are those with a GNI per capita between $1136 and $4465; upper middle-income economies are those with a GNI per capita between $4466 and $13,845; high-income economies are those with a GNI per capita of $13,846 or more” [137].

## 5. Conclusions

The elevated CMV prevalence is linked to diminished tumor occurrence across various population demographics worldwide. Combined with reduced relapse hazards evident in malignancies displaying CMV reactivation, they collectively emphasize the oncoprotective attributes of the virus. This effect is also observable in specific age intervals for a wide spectrum of tumor histologies. Supported by previous in vitro studies, these findings challenge the earlier belief that CMV acts as an etiologic factor in the manifestation of cancer and steer the prevailing opinion towards its possible oncoprotective nature. Cytomegalovirus is a complex and multifaceted virus and the ramifications following infection are far from black and white. The studies to date collectively contribute to a comprehensive understanding regarding the correlation between CMV and its oncoprotective nature. However, these conclusions warrant further scrutiny and emphasize the need for in-depth investigation to elucidate the underlying mechanisms responsible for the virus’ oncoprotective nature, while keeping in mind the epidemiological consequences influenced by the processes cited above. 

## Figures and Tables

**Figure 1 ijms-25-03741-f001:**
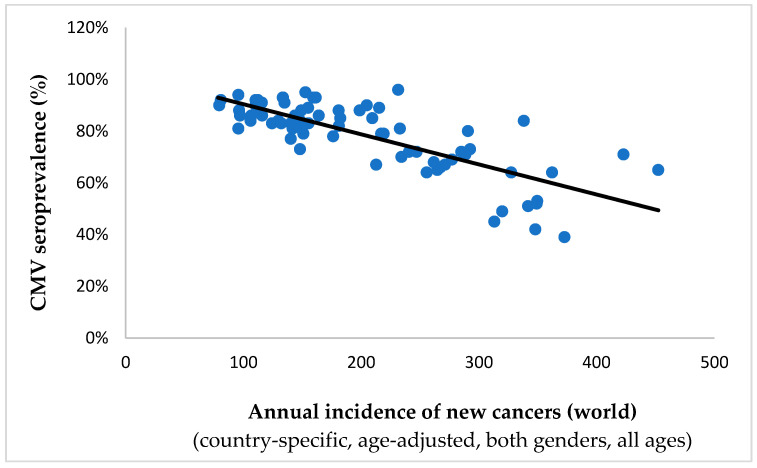
The graph represents CMV seroprevalences plotted against annual age-standardized cancer incidence rates for 73 countries (per 10^5^ persons). The statistically significant and inverse correlation between the two parameters (Spearman’s ρ = −0.732; *p* < 0.001) suggests the possible oncopreventive role of CMV: with a higher prevalence of the virus comes a lower incidence of tumors.

**Figure 2 ijms-25-03741-f002:**
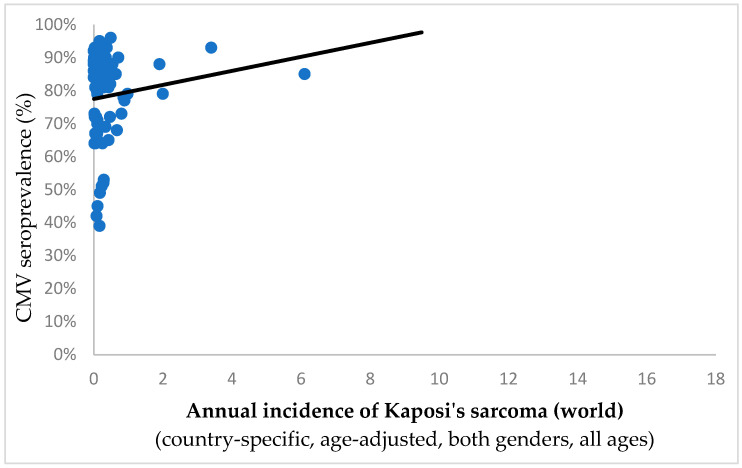
Annual incidence for Kaposi’s sarcoma (per 10^5^ persons) plotted against country specific CMV seroprevalence (Spearman’s ρ = −0.007, *p* = 0.953). In contrast to the inverse correlation noted between the pervasiveness of CMV and the overall cumulative tumor incidence rates (as seen in Figure 1), there is no discernible link in this case. This implies that CMV does not provide protection against cancer in this particular scenario.

**Figure 3 ijms-25-03741-f003:**
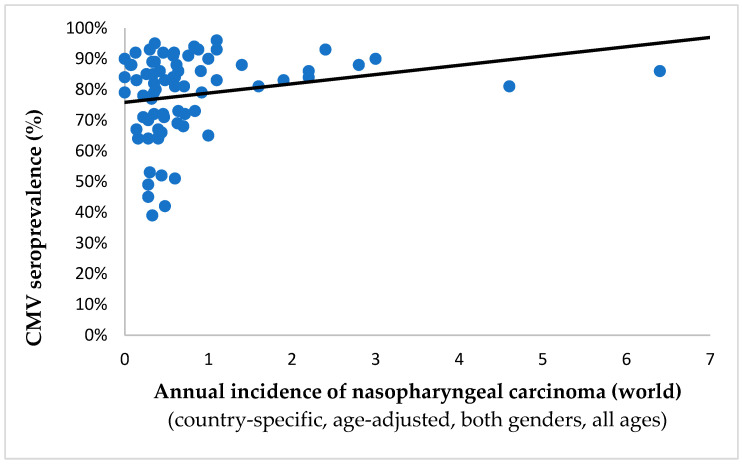
The chart illustrates the relationship between CMV seroprevalence and the incidence rates of nasopharyngeal carcinoma (per 10^5^ persons) in 73 countries across the globe. This stands out as one of just two types of cancer where a significant and positive connection has been identified at the country level (Spearman’s ρ = 0.266, *p* = 0.023), suggesting the potential carcinogenic impact of CMV in this particular case.

**Figure 4 ijms-25-03741-f004:**
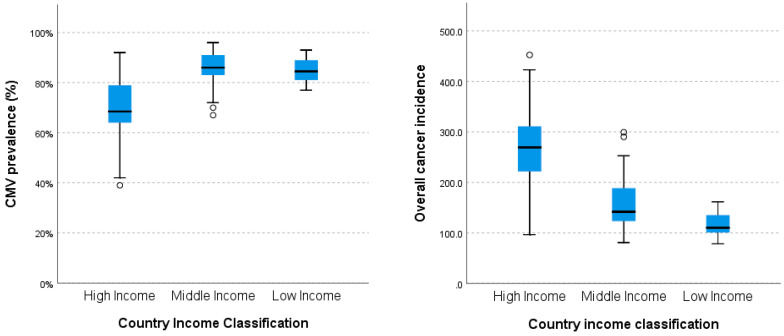
CMV prevalence and overall cancer incidence (per 10^5^ persons) according to the country’s income class. Note the lower CMV prevalences, but significantly higher cancer incidences in more affluent countries compared to their middle- and low-income counterparts.

**Table 1 ijms-25-03741-t001:** Predominant tumors as documented by the World Health Organization’s GLOBOCAN, along with their connection to global CMV prevalence, analyzed with correlation and regression analyses. Negative Spearman’s ρ and Standardized coefficients β suggest that CMV provides oncoprotection, as increased CMV prevalence is associated with decreased tumor incidence, were significant. Please note that the * sign denotes statistically significant associations.

Tumor/Localization	Correlation Analysis		Univariate Linear Regression Analysis		Multivariate Linear Regression Analysis	
	Spearman’s ρ	*p*-Value	Standardized Coefficients β	*p*-Value	Standardized Coefficients β	*p*-Value
Melanoma (skin)	−0.763	0.001 *	−0.719	<0.001 *	−0.529	<0.001 *
Kidney	−0.754	0.001 *	−0.792	<0.001 *	−0.493	<0.001 *
All cancers	−0.732	0.001 *	−0.776	<0.001 *	−0.482	<0.001 *
All cancers (excluding skin non-melanoma)	−0.726	0.001 *	−0.778	<0.001 *	−0.462	<0.001 *
Breast	−0.719	0.001 *	−0.754	<0.001 *	−0.470	<0.001 *
Testis	−0.711	0.001 *	−0.741	<0.001 *	−0.474	<0.001 *
Non-melanoma (skin)	−0.692	0.001 *	−0.497	<0.001 *	−0.372	0.006 *
Colorectum	−0.671	0.001 *	−0.665	<0.001 *	−0.280	0.001
Vulva	−0.665	0.001 *	−0.703	<0.001 *	−0.761	<0.001 *
Prostate	−0.663	0.001 *	−0.667	<0.001 *	−0.470	<0.001 *
Corpus uteri	−0.656	0.001 *	−0.610	<0.001 *	−0.306	0.006 *
Oropharynx	−0.651	0.001 *	−0.635	<0.001 *	−0.533	<0.001 *
Pancreas	−0.633	0.001 *	−0.638	<0.001 *	−0.250	0.007 *
Multiple myeloma	−0.633	0.001 *	−0.622	<0.001 *	−0.338	0.002 *
Leukemia	−0.632	0.001 *	−0.637	<0.001 *	−0.218	0.013 *
Hodgkin lymphoma	−0.618	0.001 *	−0.676	<0.001 *	−0.363	0.002 *
Non-Hodgkin lymphoma	−0.617	0.001 *	−0.624	<0.001 *	−0.336	0.001 *
Mesothelioma	−0.574	0.001 *	−0.630	<0.001 *	−0.344	0.002 *
Lip/Oral cavity	−0.551	0.001 *	−0.367	0.001 *	−0.368	0.012 *
Lung	−0.548	0.001 *	−0.573	<0.001 *	−0.259	0.011 *
Brain/CNS	−0.541	0.001 *	−0.554	<0.001 *	−0.219	0.050 *
Thyroid	−0.532	0.001 *	−0.485	<0.001 *	−0.138	0.238
Bladder	−0.519	0.001 *	−0.547	<0.001 *	−0.328	0.009 *
Ovary	−0.461	0.001 *	−0.407	<0.001 *	−0.172	0.199
Penis	−0.432	0.001 *	−0.326	0.005 *	−0.251	0.086
Hypopharynx	−0.377	0.001 *	−0.236	0.044 *	−0.261	0.084
Salivary glands	−0.35	0.002 *	−0.320	0.006 *	−0.440	0.003 *
Gallbladder	0.316	0.006 *	0.265	0.023 *	0.337	0.025 *
Nasopharynx	0.266	0.023 *	0.227	0.053	0.338	0.025 *
Vagina	−0.224	0.056	−0.105	0.377	−0.389	0.007
Larynx	−0.165	0.164	−0.117	0.325	0.016	0.913
Esophagus	−0.149	0.208	−0.052	0.660	−0.181	0.235
Cervix uteri	0.118	0.319	0.234	0.046 *	−0.175	0.166
Stomach	−0.085	0.473	0.046	0.701	0.172	0.255
Kaposi’s sarcoma	−0.007	0.953	0.135	0.255	0.031	0.836
Liver	0.007	0.951	0.185	0.117	0.202	0.184

**Table 2 ijms-25-03741-t002:** Tumor incidence rates across different age categories (in 10-year intervals) are correlated with country-specific CMV prevalence to investigate the potential oncoprotective role of CMV. We highlight *p*-values that support oncoprotection, indicating a significant and inverse correlation between viral prevalence and age-standardized tumor incidence rates, by color-coding them in green for easy reference. Conversely, *p*-values suggesting a pro-oncogenic effect are marked in red. Tumors with at least one age category indicating a potential and significant anti-tumor effect of CMV are identified with a ^†^ symbol. Note that the oncoprotective effect is somewhat skewed towards older populations in many tumors.

Tumor	Statistical Measures	Age Intervals (Years)							
		0–9	10–19	20–29	30–39	40–49	50–59	60–69	≥70
Gallbladder	Spearman’s ρ	N/A	0.154	0.297	0.394	0.416	0.423	0.290	0.147
	*p*-value	N/A	0.193	0.011	0.001	<0.001	<0.001	0.013	0.215
Bladder ^†^	Spearman’s ρ	−0.082	0.120	−0.185	−0.167	−0.254	−0.427	−0.517	−0.552
	*p*-value	0.490	0.311	0.118	0.158	0.030	<0.001	<0.001	<0.001
Colorectum ^†^	Spearman’s ρ	0.089	−0.394	−0.205	−0.373	−0.604	−0.648	−0.648	−0.662
	*p*-value	0.452	0.001	0.082	0.001	<0.001	<0.001	<0.001	<0.001
Kaposi’s sarcoma ^†^	Spearman’s ρ	−0.642	0.218	−0.083	−0.242	−0.101	0.044	0.006	0.031
	*p*-value	<0.001	0.064	0.486	0.040	0.394	0.709	0.959	0.792
Cervix uteri ^†^	Spearman’s ρ	−0.016	−0.017	−0.427	−0.220	0.036	0.249	0.373	0.400
	*p*-value	0.895	0.884	<0.001	0.061	0.761	0.034	0.001	<0.001
Corpus uteri ^†^	Spearman’s ρ	−0.196	−0.071	−0.180	−0.294	−0.576	−0.644	−0.672	−0.655
	*p*-value	0.096	0.549	0.128	0.012	<0.001	<0.001	<0.001	<0.001
Hypopharynx ^†^	Spearman’s ρ	0.050	0.205	0.307	0.187	−0.235	−0.335	−0.425	−0.378
	*p*-value	0.674	0.081	0.008	0.114	0.045	0.004	<0.001	0.001
Larynx ^†^	Spearman’s ρ	0.377	0.219	0.359	0.047	−0.103	−0.212	−0.236	−0.004
	*p*-value	0.001	0.063	0.002	0.693	0.388	0.072	0.045	0.972
Lip/Oral ^†^	Spearman’s ρ	0.375	0.244	−0.218	−0.438	−0.501	−0.536	−0.528	−0.528
	*p*-value	0.001	0.038	0.064	<0.001	<0.001	<0.001	<0.001	<0.001
Liver ^†^	Spearman’s ρ	−0.378	0.076	0.239	0.353	0.088	−0.056	−0.071	0.057
	*p*-value	0.001	0.523	0.042	0.002	0.458	0.639	0.548	0.631
Lung ^†^	Spearman’s ρ	−0.092	−0.120	−0.312	−0.308	−0.373	−0.558	−0.579	−0.501
	*p*-value	0.438	0.311	0.007	0.008	0.001	<0.001	<0.001	<0.001
Melanoma ^†^	Spearman’s ρ	−0.242	−0.692	−0.769	−0.786	−0.746	−0.749	−0.722	−0.722
	*p*-value	0.039	<0.001	<0.001	<0.001	<0.001	<0.001	<0.001	<0.001
Mesothelioma ^†^	Spearman’s ρ	0.129	−0.090	0.076	−0.235	−0.373	−0.516	−0.544	−0.595
	*p*-value	0.278	0.448	0.524	0.045	0.001	<0.001	<0.001	<0.001
Non-melanoma skin cancer ^†^	Spearman’s ρ	0.173	−0.099	−0.345	−0.411	−0.531	−0.589	−0.633	−0.709
	*p*-value	0.144	0.403	0.003	<0.001	<0.001	<0.001	<0.001	<0.001
Nasopharynx	Spearman’s ρ	0.210	0.233	0.157	0.138	0.184	0.150	0.138	0.425
	*p*-value	0.074	0.047	0.185	0.246	0.119	0.204	0.244	<0.001
Oropharynx ^†^	Spearman’s ρ	0.389	0.239	0.180	−0.423	−0.582	−0.643	−0.657	−0.611
	*p*-value	0.001	0.042	0.127	<0.001	<0.001	<0.001	<0.001	<0.001
Esophagus ^†^	Spearman’s ρ	0.298	0.406	0.208	0.192	−0.091	−0.211	−0.243	−0.025
	*p*-value	0.011	<0.001	0.077	0.104	0.445	0.073	0.039	0.833
Pancreas ^†^	Spearman’s ρ	0.052	−0.230	−0.155	−0.360	−0.594	−0.649	−0.620	−0.611
	*p*-value	0.662	0.050	0.189	0.002	<0.001	<0.001	<0.001	<0.001
Penis ^†^	Spearman’s ρ	0.171	−0.021	0.037	−0.188	−0.317	−0.337	−0.453	−0.495
	*p*-value	0.148	0.863	0.756	0.112	0.006	0.004	<0.001	<0.001
Prostate ^†^	Spearman’s ρ	−0.058	0.029	0.185	−0.126	−0.602	−0.663	−0.708	−0.542
	*p*-value	0.628	0.807	0.118	0.289	<0.001	<0.001	<0.001	<0.001
Salivary glands ^†^	Spearman’s ρ	0.198	−0.078	−0.226	−0.276	−0.417	−0.359	−0.126	−0.429
	*p*-value	0.093	0.509	0.055	0.018	<0.001	0.002	0.287	<0.001
Testis ^†^	Spearman’s ρ	−0.319	−0.653	−0.694	−0.707	−0.728	−0.728	−0.486	−0.097
	*p*-value	0.006	<0.001	<0.001	<0.001	<0.001	<0.001	<0.001	0.415
Thyroid ^†^	Spearman’s ρ	−0.266	−0.489	−0.467	−0.491	−0.499	−0.550	−0.544	−0.363
	*p*-value	0.023	<0.001	<0.001	<0.001	<0.001	<0.001	<0.001	0.002
Vulva ^†^	Spearman’s ρ	0.044	−0.003	−0.192	−0.384	−0.549	−0.544	−0.642	−0.745
	*p*-value	0.709	0.980	0.103	0.001	<0.001	<0.001	<0.001	<0.001
All cancers ^†^	Spearman’s ρ	−0.642	−0.672	−0.731	−0.784	−0.756	−0.726	−0.720	−0.678
	*p*-value	<0.001	<0.001	<0.001	<0.001	<0.001	<0.001	<0.001	<0.001
Vagina ^†^	Spearman’s ρ	−0.033	0.126	<0.001	−0.125	−0.257	−0.103	−0.189	−0.376
	*p*-value	0.785	0.287	0.999	0.291	0.028	0.385	0.110	0.001
Stomach	Spearman’s ρ	0.283	0.145	0.079	0.039	−0.042	−0.055	−0.081	−0.097
	*p*-value	0.015	0.221	0.504	0.746	0.723	0.644	0.496	0.413
Ovary ^†^	Spearman’s ρ	−0.121	−0.181	−0.071	−0.188	−0.284	−0.395	−0.517	−0.454
	*p*-value	0.308	0.124	0.551	0.111	0.015	0.001	<0.001	<0.001
Brain/CNS ^†^	Spearman’s ρ	−0.649	−0.577	−0.654	−0.607	−0.529	−0.477	−0.455	−0.452
	*p*-value	<0.001	<0.001	<0.001	<0.001	<0.001	<0.001	<0.001	<0.001
All cancers excl. non-melanoma skin ^†^	Spearman’s ρ	−0.645	−0.667	−0.733	−0.785	−0.753	−0.726	−0.717	−0.656
	*p*-value	<0.001	<0.001	<0.001	<0.001	<0.001	<0.001	<0.001	<0.001
Breast ^†^	Spearman’s ρ	0.050	0.182	−0.239	−0.645	−0.674	−0.696	−0.700	−0.661
	*p*-value	0.674	0.124	0.042	<0.001	<0.001	<0.001	<0.001	<0.001
Kidney ^†^	Spearman’s ρ	−0.493	−0.016	−0.365	−0.693	−0.736	−0.731	−0.728	−0.744
	*p*-value	<0.001	0.893	0.002	<0.001	<0.001	<0.001	<0.001	<0.001
Hodgkin lymphoma ^†^	Spearman’s ρ	0.174	−0.622	−0.676	−0.681	−0.575	−0.478	−0.389	−0.229
	*p*-value	0.141	<0.001	<0.001	<0.001	<0.001	<0.001	0.001	0.051
Non-Hodgkin lymphoma ^†^	Spearman’s ρ	−0.036	−0.005	−0.437	−0.590	−0.593	−0.618	−0.610	−0.552
	*p*-value	0.762	0.968	<0.001	<0.001	<0.001	<0.001	<0.001	<0.001
Multiple myeloma ^†^	Spearman’s ρ	0.235	0.110	0.055	−0.356	−0.483	−0.577	−0.644	−0.627
	*p*-value	0.045	0.355	0.645	00.002	<0.001	<0.001	<0.001	<0.001
Leukemia ^†^	Spearman’s ρ	−0.495	−0.236	0.108	−0.165	−0.472	−0.630	−0.645	−0.543
	*p*-value	<0.001	0.044	0.364	0.164	<0.001	<0.001	<0.001	<0.001

## Data Availability

All data are either publicly accessible or available from the corresponding author upon reasonable request.

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
