# Peer review of "Human Cytomegalovirus Oncoprotection across Diverse Populations, Tumor Histologies, and Age Groups: The Relevance for Prospective Vaccinal Therapy"

_ijms, 2024, doi:10.3390/ijms25073741_

Round 1
Reviewer 1 Report
Comments and Suggestions for Authors
Specific comments:
1. Figure 4 is blurry. Please replace it with higher quality graphs. Please also clarify the cutoff or criteria for high/middle/low income countries either in the legend or methods section.
2. In Figures 2, 3, and 4, please specify the unit for cancer incidence. Is it per 100,000 population?
3. All figures: Y-axis titles say CMV seroprevalence (%) while the values are all under 1. Please correct this.
4. Some references are out of order, so they need to be reordered. For example, reference 14 first appears in the methods section (line 445), and reference 129-131 (lines 243-244) appears in between references 30 and 31.
5. Discussion section 3.1 (lines 271 to 305: The section needs rewriting. There are multiple issues here.
First, the authors seem to conflate anti-cancer protection offered by natural CMV infection with various CMV-based anti-cancer therapies being studied/developed. Using CMV as a biomarker or targeting CMV as an anticancer therapy for cancers that have CMV infected cells is different than using CMV as a therapeutic tool for cancers that DONOT have CMV infected cells. Please clarify between these scenarios when discussing the utility of CMV-based therapies.
Second, various CMV-based therapies, such as CMV-based vaccine vectors, use of CMV promoter for gene therapy, and CMV as an anti-cancer vaccine are all presented without much clarity on how these differ or where they might be useful.
Third, at times, the authors seem to overinterpret the conclusion from the cited references in this section. For example, the authors talk about using CMV as an anticancer vaccine for gliomas (lines 288-296), and suggest that this could be a useful strategy for all different types of cancers. It seems disingenuous to leave out of the discussion that CMV antigens and DNA are present at a higher rate in glioblastomas and that some studies suggest CMV might contribute to the pathogenesis of this cancer.
6. It is interesting that the authors find no correlation between CMV and KS. The authors hypothesize that the attenuated T-cell response in KS patients (due to HIV/AIDS) might be the reason for this. It would be informative to look at the seroprevalence of CMV in countries where classical KS is prevalent and assess how that compares to the overall trend in rest of the countries. Do countries with classical KS have particularly low (or high) CMV prevalence?
7. Since NPC is only one of 2 cancers showing a positive association with CMV seroprevalence, it would be helpful to further elucidate whether EBV status affects this correlation.
Author Response
Point-by-point response to the Reviewer 1
(March 11, 2024)
Journal: International Journal of Molecular Sciences
Manuscript title: Human Cytomegalovirus Oncoprotection Across Diverse Populations, Tumor Histologies, and Age Groups: The Relevance for Prospective Vaccinal Therapy
Manuscript ID: ijms-2876029
Reviewer 1 Comments
Reviewer 1, comment 1:
Specific comments:
- Figure 4 is blurry. Please replace it with higher quality graphs. Please also clarify the cutoff or criteria for high/middle/low income countries either in the legend or methods section.
Authors’ reply to comment 1:
Foremostly, dedicated Reviewer 1 evaluation our manuscript is truly appreciated as well as her constructive feedback enhances the quality of early draft.
We have amended the quality of Figure 4. Regarding significance of the income level, relevant remarks were added in Materials and Methods: “…to inquire into the country income level, data from the World Bank was used [139]”. The World Bank provides a purvey of respective income categories: “For the current 2024 fiscal year, low-income economies are defined as those with a GNI per capita, calculated using the World Bank Atlas method, of $1,135 or less in 2022; lower middle-income economies are those with a GNI per capita between $1,136 and $4,465; upper middle-income economies are those with a GNI per capita between $4,466 and $13,845; high-income economies are those with a GNI per capita of $13,846 or more.” This is now added to Materials and Methods, lines 479 to 485.
Reviewer 1, comment 2:
- In Figures 2, 3, and 4, please specify the unit for cancer incidence. Is it per 100,000 (/105) population?
Authors’ reply to comment 2:
We added (per 105 persons), specifying adequate parameters of incidence.
Reviewer 1, comment 3:
- All figures: Y-axis titles say CMV seroprevalence (%) while the values are all under 1. Please correct this.
Authors’ reply to comment 3:
We duly made corrections you required. The Y-axis now represents CMV percentage values (0 to 100%).
Reviewer 1, comment 4:
- Some references are out of order, so they need to be reordered. For example, reference 14 first appears in the methods section (line 445), and reference 129-131 (lines 243-244) appears in between references 30 and 31.
Authors’ reply to comment 4:
Indeed, we corrected the order of the references. How stupid of our service!
Reviewer 1, comment 5:
- Discussion section 3.1 (lines 271 to 305: The section needs rewriting. There are multiple issues here.
First, the authors seem to conflate anti-cancer protection offered by natural CMV infection with various CMV-based anti-cancer therapies being studied/developed. Using CMV as a biomarker or targeting CMV as an anticancer therapy for cancers that have CMV infected cells is different than using CMV as a therapeutic tool for cancers that DONOT have CMV infected cells. Please clarify between these scenarios when discussing the utility of CMV-based therapies.
Second, various CMV-based therapies, such as CMV-based vaccine vectors, use of CMV promoter for gene therapy, and CMV as an anti-cancer vaccine are all presented without much clarity on how these differ or where they might be useful.
Third, at times, the authors seem to overinterpret the conclusion from the cited references in this section. For example, the authors talk about using CMV as an anticancer vaccine for gliomas (lines 288-296), and suggest that this could be a useful strategy for all different types of cancers. It seems disingenuous to leave out of the discussion that CMV antigens and DNA are present at a higher rate in glioblastomas and that some studies suggest CMV might contribute to the pathogenesis of this cancer.
Authors’ reply to comment 5:
Thank you, Reviewer 1, for bringing our attention to this matter. Understanding the intricate nature of the CMV's role is crucial, and our utmost desire is to effectively and comprehensively tackle this issue, expounding on it with clear and concise prose.
The reason why we heavily rely on referencing CMV-based anti-cancer therapies is because of the (purportedly) same mechanism that underlies CMV oncoprevention and these therapeutic regimens – the CMV-galvanized T/NK-cell anti-tumor response. These studies serve as an in vitro argument in strengthening the validity of our serology-based line of reasoning. Simply put, what may be interpreted as "conflating" is, in truth, our intention to provide an outlook on the similar effects of CMV primoinfection and CMV-based therapies within the study’s narrative.
However, we have indeed removed the parts of the text dealing with CMV as a vector for tumor antigens, as it is not pertinent for this study, as aptly observed by the Reviewer 1 – thank you!
Finally, the Reviewer's observation regarding our potential overinterpretation of conclusions from glioblastoma studies and (invalid) extrapolation to other malignancies is quite sound and valid. Consequently, we have modified our Discussion to include a cautionary note on this matter.
We have now tried to the best of our abilities to mend the text according to Reviewer 1’s astute suggestions, and we hope that the text is clearer to the reader, transmitting the proper message and scientific rationale we aimed for. We would very much appreciate it if Reviewer 1 could revisit this section and share their perspective with us. The revised and, hopefully, more lucid and precise text can be found from line 271 to 302.
Reviewer 1, comment 6:
- It is interesting that the authors find no correlation between CMV and KS. The authors hypothesize that the attenuated T-cell response in KS patients (due to HIV/AIDS) might be the reason for this. It would be informative to look at the seroprevalence of CMV in countries where classical KS is prevalent and assess how that compares to the overall trend in rest of the countries. Do countries with classical KS have particularly low (or high) CMV prevalence?
Authors’ reply to comment 6:
This lack of correlation is an indirect evidence for a T-cell mediated anti-tumor activity based on CMV antigens – being a current opinion in, say, (Cuburu et al., PNAS, 2022). Classical KS typically occurs in middle-aged or older adults without HIV, and is often diagnosed in men of Mediterranean, eastern European, and Middle Eastern heritage. Regrettably, we do not possess the disaggregated values for each of the KS types, that would delineate between any of the four categories of KS (as defined by their epidemiological features). We would point to the Reviewer’s recent publication Fu et al. (Lancet, 2023: Global patterns and trends in Kaposi sarcoma incidence: a population-based study), where the KS incidence was, it seems, observed jointly. Finally, as the WHO ICD coding system for diseases (the International Classification of Diseases: https://icd.who.int/browse/2024-01/mms/en#2131977134) lists only “Kaposi’s sarcoma”, not delineating between Classical/Endemic/Iatrogenic, we could hardly attain to data analysis for a separate KS type.
Reviewer 1, comment 7:
- Since NPC is only one of 2 cancers showing a positive association with CMV seroprevalence, it would be helpful to further elucidate whether EBV status affects this correlation.
Authors’ reply to comment 7:
The presence of EBV is a confounding factor in ethology of nasopharyngeal carcinoma, a worthy fact suggested by the Reviewer 2. We overlooked mentioning it in an initial draft. Reviewer 1 spotted this important point. Investigating the co-infection with several Orthoherpesviruses would certainly further elaborate on the pathogen-tumor relationship. We analysed association of EBV as a separate variables but had several hurdles to this:
- EBV permeates the majority of the world’s population, and by a young age; it would be difficult (if not impossible) to obtain a country-by-country seroprevalence of EBV which encompasses older EBV-free individuals to further compare for EBV as a confounder in this case (Dowd JB, et al. PLoS One. 2013);
- Literature is, unfortunately, scant considering EBV+CMV co-infections as regards entire country populations; we are afraid of yielding spurious data if dealing lesser numbers of subjects (e.g. single-center and similar studies). Having above-mentioned in mind, we placed a short paragraph into the “3.4. Study Limitations” elaborates on a possible confounding effect of EBV in this case (lines 424 to 427).
Reviewer 2 Report
Comments and Suggestions for Authors
This study investigated CMV seroprevalence in 73 countries and its correlation with the age-standardized incidence rate of invasive tumors. The study took into account the possible age at first infection and used the database of the International Agency for Research on Cancer of the World Health Organization.
The results show a possible inverse correlation between the presence of antibodies (and perhaps infection) and tumor development, but this is not the case for all conditions and for some tumors, particularly those known to be permissive for CMV replication.
The work cannot and will not draw definitive conclusions. There are too many variables that were not considered that could have led to different results. The paper seems to imply that the immune response elicited against CMV is protective against the tumor. The authors need to provide more details about the study population. If they are immunocompromised individuals, the protective effect of the anti-CMV immune response may not be present. In addition, the authors need to check whether the study population includes patients undergoing anti-CMV treatment, as they may have a reduced anti-CMV immune response, especially if treatment is prolonged. Authors must provide this information and consider and discuss these variables if they wish to publish their paper.
Comments on the Quality of English LanguagePlease check abbreviations for consistency.
Author Response
Authors’ reply to the Reviewer 2:
Journal: International Journal of Molecular Sciences
Manuscript title: Human Cytomegalovirus Oncoprotection Across Diverse Populations, Tumor Histologies, and Age Groups: The Relevance for Prospective Vaccinal Therapy
Manuscript ID: ijms-2876029
Thank you very much for kindly pointing out important hints to us. We find them significant and discuss them below in some detail.
Our current work expands on our earlier single-centre clinical study on B lymphocyte malignancies [Janković et al. Virol J. 19(1):155. 2022; doi: 10.1186/s12985-022-01884-1; Ed. Wolfram Brune]. In it we demonstrated a protective effect conferred by CMV infection on its human host. Apart from our present work, there has been no comprehensive or global exploration regarding a possible time-dependent oncogenesis (or oncoprotection) linked with CMV infection, as yet.
Most recently, three elaborate clinical studies came forth (please, see the refs. below on colorectal cancer, bronchogenic carcinoma, and multiple myeloma). They seem to support both our initial (Virol J 2022) and presently submitted evidence at a global level:
▪ Nagel B et al. The Association of Human Cytomegalovirus Infection and Colorectal Cancer: A Clinical Analysis. World J Oncol. 2023;14(2):119-124. doi: 10.14740/wjon1565.
▪ Rashid S et al. Human Cytomegalovirus (CMV) Infection Associated With Decreased Risk of Bronchogenic Carcinoma: Understanding How a Previous CMV Infection Leads to an Enhanced Immune Response Against Malignancy. Cureus. 2023;15(4):e37265. doi: 10.7759/cureus.37265.
▪ Salwender H et al. Cytomegalovirus immunoglobulin serology prevalence in patients with newly diagnosed multiple myeloma treated within the GMMG-MM5 phase III trial. Hematology. 2024; 29(1), 2320006. doi.org/10.1080/16078454.2024.2320006
Our work correlates decadal distribution of the age-standardized incidence rate of invasive tumors (the database of the International Agency for Research on Cancer of the World Health Organization) with the country-specific IgG antibody seroactivity rates of CMV. It was Lehrer who was the first to notice that the age at infection may be important in an anti-cancer or prooncogenic (risk) effect of CMV on glioblastoma [Lehrer S. Cytomegalovirus infection in early childhood may be protective against glioblastoma multiforme, while later infection is a risk factor. Medical hypotheses. 2012; 78(5), 657-658. doi.org/10.1016/j.mehy.2012.02.003.].
Our study is suggestive albeit, of course, cannot be entirely confirmatory regarding an anti- or potentially pro-oncogenic effect of CMV in terms of age strata at which infection (i.e. IgG seropositivity) has occurred and regarding cancer histologies of interest. So, again, the Referee 2 is very much correct that no definitive conclusions could be drawn from this work. Definitive conclusion, of course, not – but cautious and explorative interpretation may be worth publishing in the all-important context of oncological virology.
The data (and references) on study populations, commented on by the Referee 2, were again precisely detailed and provided by an extensive Appendix to Zuhair et al. [Estimation of the worldwide seroprevalence of cytomegalovirus: A systematic review and meta‐analysis. Rev Med Virol. 2019;e2034. doi: 10.1002/rmv.2034]. Appendix is freely available on-line but we shall have quickly forwarded it to this Reviewer if need be. Zuhair et al. relied on the very respected national health and nutrition examination survey (NHANES) and on extensive literatural data on country-specific immunoglobulin G serology with a large geographical coverage. They provide data availability in terms of site-years.
Perceptively enough, the Referee 2 observes that in the immunocompromised individuals (for example, post-chemotherapy patients and Kaposi sarcoma patients in particular) the protective effect of the anti-CMV immune response may not be present. How so very true! We have studied (considering it sort of a “control group”) the worldwide population of patients with Kaposi sarcoma (GLOBOCAN database of WHO) and have, indeed, found no protective effect of CMV in these individuals – as correctly predicted by the Referee 2. This work is currently being under consideration for publication (in Virol J) but we shall be delighted to immediately forward it to this Referee for her inspection (the data regarding the population of Kaposi sarcoma patients is recorded in Fig. 5 of that work). Of importance, we were careful to check whether the study population includes patients undergoing anti-CMV treatment, as they may have a reduced anti-CMV immune response, especially if treatment is prolonged. No only newly diagnosed patients were included as was also the case with the most recent report (Hematology; 2024) on CMV protection against multiple myeloma (please, view Salwender H et al. above).
As, we wish to publish this work, and perhaps this Referee may also wish that, we discuss in the relevant section (Discussion) the importance of very many variables (encompassed as the socio-demographic index (SDI) or socio-economic status (SES). SES predicts local odds of CMV seropositivity due to poverty and SDI summarily measures a mean estimate of a location’s position on a spectrum of development. Indeed, poverty (with its prevailing CMV seropositivity) seems to protect against cancer and it inversely correlated with the incidence of cancers. SDI is a composite indicator that includes fertility, education, and income. We certainly considered the possibility that some other variables may have influenced our evidence that implies CMV as protective against tumors. A comprehensive statistical analysis of data across the global landscape was performed (kindly view the Supplementary material) but some risk profiles, that are also imbalanced, would always remain elusive.
We cherish some hope that, after due corrections in the MS, the Referee 2 may find the current work worthwhile appearing in print in this Journal.
Round 2
Reviewer 1 Report
Comments and Suggestions for Authors
The authors have adequately addressed my comments. My only minor comment is:
1. Fix Figure 2 Y-axis so that the best-fit line is not cut-off abruptly.
Author Response
Reviewer 1 comment:
Fix Figure 2 Y-axis so that the best-fit line is not cut-off abruptly.
Authors' answer:
We have now adjusted the Figure 2 so that the Trend-line does not end suddenly, appearing sharply cut off from the graph.
I would also like to express my gratitude once again to Reviewer 1 for their constructive feedback, which has significantly enhanced the quality of our manuscript. Additionally, I appreciate their kind and constructive communication throughout this process.
Reviewer 2 Report
Comments and Suggestions for Authors
The authors properly addressed all raised issues. The article is suitable for publication in IJMS
Author Response
Reviewer 2's comment:
Authors' reply:
I would like to express my gratitude once again to Reviewer 2 for their feedback, which has significantly enhanced the quality of our manuscript. Additionally, I appreciate their kind and constructive communication throughout this process.
It is always gratifying and reassuring for the authors involved to experience a high-quality interaction with the attending reviewers. My personal sincerest best wishes and kind regards to the Reviewer 2.